# Arginine Supply Impacts the Expression of Candidate microRNA Controlling Milk Casein Yield in Bovine Mammary Tissue

**DOI:** 10.3390/ani10050797

**Published:** 2020-05-05

**Authors:** Xin Zhang, Yifan Wang, Mengzhi Wang, Gang Zhou, Lianmin Chen, Luoyang Ding, Dengpan Bu, Juan Loor

**Affiliations:** 1State Key Laboratory of Animal Nutrition, Institute of Animal Science, Chinese Academy of Agricultural Sciences, No. 2 Yuanmingyuan West Road, Beijing 100193, China; kakashi2012@163.com; 2College of Animal Science and Technology, Yangzhou University, 88 South University Ave., Yangzhou 225009, China; yzdxzg@163.com (G.Z.); lianmin.chen@rug.nl (L.C.); luoyang.ding@research.uwa.edu.au (L.D.); 3School of Clinical Medicine, Southeast University, 87 Dingjiaqiao Road, Nanjing 210009, China; gilmore2333@163.com; 4Department of Animal Sciences and Division of Nutritional Sciences, University of Illinois, 1207 W. Gregory Drive, Urbana, IL 61801, USA; jloor@illinois.edu

**Keywords:** arginine, casein yield, microRNAs, milk, gene expression regulation

## Abstract

**Simple Summary:**

It has been reported that arginine plays an important role in lactation, including promoting mammary gland development, increasing yields of milk and casein. Recent studies revealed that microRNA could be involved in regulating expression of functional genes related to mammary development. Thus, exploring the role of arginine on the regulation of miRNA related to bovine mammary development and milk production in dairy cows is of importance. The present work revealed that arginine injection increased casein yield by altering the expression of selected microRNA associated with mammary development.

**Abstract:**

Arginine, a semi-essential functional amino acid, has been found to promote the synthesis of casein in mammary epithelial cells to some extent. Data from mouse indicated that microRNA (miRNA) are important in regulating the development of mammary gland and milk protein synthesis. Whether there are potential links among arginine, miRNA and casein synthesis in bovine mammary gland is uncertain. The objective of the present work was to detect the effects of arginine supplementation on the expression of miRNA associated with casein synthesis in mammary tissue and mammary epithelial cells (BMEC). The first study with bovine mammary epithelial cells (BMEC) focused on screening for miRNA candidates associated with the regulation of casein production by arginine. The BMEC were cultured with three different media, containing 0, 1.6 and 3.2 mM arginine, for 24 h. The expression of candidate miRNA was evaluated. Subsequently, in an in vivo study, 6 Chinese Holstein dairy cows with similar BW (mean ± SE) (512.0 ± 19.6 kg), parity (3), BCS (4.0) and DIM (190 ± 10.3 d) were randomly assigned to three experimental groups. The experimental cows received an infusion of casein, arginine (casein plus double the concentration of arginine in casein), and alanine (casein plus alanine, i.e., iso-nitrogenous to the arginine group) in a replicated 3 × 3 Latin square design with 22 d for each period (7 d for infusion and 15 d for washout). Mammary gland biopsies were obtained from each cow at the end of each infusion period. Results of the in vitro study showed differences between experimental groups and the control group for the expression of nine miRNA: miR-743a, miR-543, miR-101a, miR-760-3p, miR-1954, miR-712, miR-574-5p, miR-468 and miR-875-3p. The in vivo study showed that arginine infusion promoted milk protein content, casein yield and the expression of *CSN1S1* and *CSN1S2*. Furthermore, the expression of miR-743a, miR-543, miR-101a, miR-760-3p, miR-1954, and miR-712 was also greater in response to arginine injection compared with the control or alanine group. Overall, results both in vivo and in vitro revealed that arginine might partly influence casein yield by altering the expression of 6 miRNAs (miR-743a, miR-543, miR-101a, miR-760-3p, miR-1954, and miR-712).

## 1. Introduction

Arginine, a semi-essential functional amino acid, plays an important role in animal nutrition. It has been shown that the addition of arginine to diets effectively increases average daily weight gain and improves the growth performance of animals [1]. Furthermore, some studies also indicated that arginine plays an important role in milk protein synthesis in mammals. For example, Chew et al. [2] found that arginine infusion in late-pregnant cows dramatically increased concentrations of prolactin, growth hormone, and insulin in blood and subsequently increased milk yield. Furthermore, the study by Ma et al. [3] indicated that the supplementation of arginine stimulated protein synthesis and reduced protein degradation in porcine mammary epithelial cells, leading to increases in cell proliferation and the production of major milk proteins. A recent study in lactating cows also found that a deficiency of arginine had a negative effect on milk protein production [4]. In other studies, the effects of arginine on protein synthesis have been attributed to metabolites of arginine, such as nitric oxide and polyamines, which regulate the growth of mammary gland and blood vessels [5,6]. A study in bovine mammary epithelial cells (BMEC) also found that the supplementation of arginine promoted the synthesis of casein and expression of *CSN1S1*, *CSN1S2*, *CSN2*, and *CSN3* by regulating the signaling pathway of tyrosine kinase 2-signal transduction and transcriptional activator 5 (JAK2-STAT5) and rapamycin target protein (mTOR) [7].

Studies in dairy cows and rats have revealed that the expression profiles of selected miRNA differ across stages of lactation, underscoring the potential importance of these molecules in milk production [8,9]. A previous study in mice revealed that the expression of miR-431, which is responsible for the sparse spiral ganglion neurons by suppressing EYA4 (Eyes absent homolog 4) translation, was down-regulated as development progressed [10]. However, the expression of miR133, which enhances myoblast proliferation by repressing the serum response factor [11], was up-regulated during pregnancy and lactation. These results suggested that those miRNAs are functionally involved in murine mammary gland development. Murine mammary cell proliferation and β-casein gene expression are partly regulated by miR-101a [12]. In the context of milk components, data indicate that the 3’UTR of the casein genes (CSN1S1, CSN1S2, CSN2, and CSN3) in goats is located in the complementary regions of various predicted miRNA seed sequences [13].

Taking into consideration the results of previous studies, it is possible that the changes in the expression of casein genes and related functional genes that respond to increased arginine concentration are modulated by miRNA. Thus, the objective of this work was to uncover the miRNAs involved in the regulation of milk protein production induced by increased arginine supply.

## 2. Materials and Methods

### 2.1. In Vitro Experiment

#### 2.1.1. Experimental Design and Culture Medium

The methods for the in vitro experiment were similar to those described by Wang et al. [14] Briefly, bovine mammary epithelial cells (BMEC) were cultured for 24 h in three different media containing 0 (Control group), 1.60 (Casein group), and 3.20 mM (Arginine group) arginine. The arginine-containing medium (Table 1) had a profile of amino acids similar to casein [15]. In brief, the basic medium devoid of arginine was prepared by combining the individual reagents including vitamins, minerals, and amino acids in ultrapure water (according to the composition of DMEM/F12, Gibco, Invitrogen, Catalog #11320082, Life Technologies Corporation, Burlingont, ONT, Canada). The medium pH was adjusted to 7.2 with NaHCO_3_. All reagents used in cell culture were purchased from Sigma Chemical Co. (St. Louis, MO, USA). Before the arginine study, the basic medium was used in the cell resuscitation process. 

#### 2.1.2. Cell Resuscitation and Treatment

The cell resuscitation process was described in previous papers [7,14]. In brief, second-generation mammary epithelial cells were taken from liquid nitrogen and placed in a 37 °C water bath. Cells were centrifuged at 150× *g* for 5 min at 4 °C to isolate BMEC, and the precipitate rinsed with fresh medium prior to transferring to a new culture bottle. Once 90% confluence was reached, cultured cells were harvested with 0.25% trypsin EDTA and seeded at a density of 5 × 10^4^ cells/mL into 6-well plates (3335, Corning Life Science, New York, NY, USA), containing growth medium (DMEM/F12 with 10% Fetal bovine serum (FBS (16000-044, Gibco, Carlsbad, CA, USA)), 500 ng/mL hydrocortisone (S0135, Sigma, Saint Louis, MI, USA), 1 µg/L prolactin (L6520, Sigma, Saint Louis, MI, USA), 10 ng/mL epidermal growth factor (AF-100-15, Peprotech, Rocky Hill, CT, USa), and 100 IU/mL penicillin/streptomycin)). Each treatment was replicated 6 times and cultured in an incubator at 37 °C, 95% O_2_, and 5% CO_2_. Cells were incubated with DMEM/F12 medium without fetal bovine serum for 16 h, to eliminate the effects from serum, then the cells were incubated in medium with different treatments for another 24 h.

#### 2.1.3. Database Prediction and Result Acquisition

The prediction of miRNA was carried out with PicTar (http://pictar.mdc-berlin.de/) and the TargetScan (http://www.targetscan.org/) database. The first step was to identify highly-conserved miRNA candidates with high context scores, by searching target genes in both PicTar and TargetScan databases. Secondly, the miRNA candidates from both PicTar and TargetScan were listed in the same Excel file, and the miRNA overlapping both databases were selected. Lastly, we selected the miRNA which regulate multiple target genes (at least 2) for further analysis (Table 2). 

#### 2.1.4. Detection of miRNA Expression by RT-qPCR

The miRNA were isolated from cells with a miRNA Isolation Kit according to the manufacturer’s protocols (Tian Gen, Catalog # DP501, Beijing, China). The miRNA were polyadenylated using Poly (A) polymerase, according to the manufacturer’s instructions (miRcute miRNA First-Strand Synthesis Kit, Tian Gen, Catalog # KR201, Beijing, China). cDNA was synthesized with Oligo (dT)—universal tag and reverse transcription primer. RT-PCR analysis was done using the SYBR Green method with the miRcute miRNA qPCR Detection Kit (Tian Gen, Catalog # FP401, Beijing, China) in the ABI Prism 7500 Detection Instrument (Applied Biosystems, Foster, CA, USA). Each sample was run in triplicate. The internal reference gene was U6 [16], and the primer sequence from 5′ to 3′ was F—CGC TTC GGC AGC ACA TAT AC & R—TTC ACG AAT TTG CGT GTC AT. The expression of U6 was not altered by treatment. The other primer sequences were the complete miRNA sequences.

### 2.2. In Vivo Experiment

#### 2.2.1. Experimental Animals and Management

Six lactating Holstein cows with similar BW (mean ± SE) (512.0 ± 19.6 kg), parity (3), BCS (4.0) and DIM (190 ± 10.3 d) were selected for this experiment from the Experimental Farm of Yangzhou University. All animal procedures were approved by the Yangzhou University Animal Care and Use Committee of Jiangsu Province (China).

The basal diet was formulated according to NRC [17]. The diet composition is reported in Table 3. Cows were fed twice a day at 0600 and 2000 h, and feed refusals collected to calculate daily DM intake. The experimental cows were milked 3 times a day at 0700, 1500 and 2300 h. Cows were housed separately in a free-stall barn and had ad libitum access to the TMR and freshwater. Indwelling catheters (137 mm: 1.2 mm i.d.: 2.0 mm o.d., L13712, Jiangxi Huali Medical Instrument Company, Ganzhou, China) were placed in a jugular vein and flushed with heparin and physiological saline (750 IU/mL) twice daily during the week before infusion treatments.

#### 2.2.2. Experimental Design

Six experimental animals were randomly divided into 3 treatments (2 cows per group) in a replicated 3 × 3 Latin square design with 22 days for each period (day 1 to day 7 for infusion and day 8 to day 22 for washout). The treatments were casein group (casein model), arginine group (casein plus double the concentration of arginine in casein), and alanine group (added alanine to make the it iso-nitrogenous to arginine group). The perfusates (Table 4) were manufactured by Cambridge Biological Company (Nanjing, China), and infused continuously through a peristaltic pump (Longer, Hebei, China) for 8 h/d (from 0600 to 1400 h). The total perfusion for each cow was 4 L in 8 h.

#### 2.2.3. Samples and Data Collection

Milk yield during infusion was recorded every day, and milk samples taken thrice daily at 0700, 1500 and 2300 h and mixed to create a pool in proportion to milk yield at each milking. A subsample of the milk was used for measuring protein, fat, and non-fat milk solids (Bentley FTS/FCM 400 Combi; Bentley Instrument Inc., Chaska, MN, USA). Another subsample was centrifuged at 2810× *g* at 4 °C for 10 min to separate milk fat, for the determination of casein concentrations in milk. The concentrations of α-casein, β-casein and κ-casein were detected using the bovine ELISA kits from Cloud-Clone Corp. (SEJ333Bo, SEJ332Bo, and SEJ331Bo, respectively, Cloud-Clone Corp., Houston, TX, USA).

At the end of each infusion period, around 200 mg mammary gland tissue were taken from each cow using a published biopsy method [18] and placed in liquid N prior to storage at −80 °C until RNA extraction. Frozen mammary tissue was quickly minced and immediately subjected to RNA extraction with ice-cold TRIzol (15596018, ThermoFisher, Carlsbad, CA, USA) as described previously, including a DNAse digestion step [19]. The RNA integrity was assessed via electrophoresis analysis and the RNA concentration was measured with a Nanodrop spectrophotometer (ThermoFisher). The RNA was diluted to 100 ng/µL before reverse-transcription and cDNA synthesis with the High-Capacity cDNA Reverse Transcription Kit (4368813, Applied Biosystems, Waltham, MA, USA). 

Subsequently, the qRT-PCR analysis of target genes screened out in the in vitro experiment (Table 5) was performed using the Power SYBR Green PCR Master Mix (4367659, Applied Biosystems). Each sample was run in triplicate and the reactions were performed in the ABI Prism 7500 Detection Instrument (Applied Biosystems) using the protocol below: 30 s at 95 °C, 10 s at 95 °C, 20 s annealing temperature, and 30 s at 72 °C for 40 cycles. The same conditions were used on an equal amount of RNAse-free water as a negative control. Primers were synthesized by Biotech Bioengineering Co. Ltd. (Shanghai, China). Glyceraldehyde-3-phosphate dehydrogenase (GAPDH) and β-actin (ACTB) were selected as internal control genes. The expression of GAPDH and ACTB was not altered by treatments. 

### 2.3. Statistical Analysis

The mRNA expression was assessed with the 2^-ΔΔCt^ method [20], where ΔΔCt = (Ct _target gene_ of the experimental group − Ct _internal control_ of experimental group) − (Ct _target gene_ of the control group − Ct _internal control_ of the control group). The statistical analysis was carried out with SPSS (v16.0). Variance homogeneity and normality tests were performed, and the 4 variables (miRNA-181d, miRNA-329, miR-3062, α-Casein) were transformed using the square root transformation to normalize distributions prior to analysis. A one-way ANOVA was used to test differences among groups, while Tukey’s multiple comparison test was used to determine differences between treatments. The fixed effect considered in the model was the treatment, and cow represented the random effect. Significance was declared at *p* < 0.05. The same statistical approach was used for both in vitro and in vivo data.

## 3. Results

### 3.1. miRNA Expression in Mammary Cells In Vivo 

The expression of detectable miRNA was reported in Table 6. Only 15 of the 53 miRNA screened from the casein synthesis pathway by database prediction were detectable. Among those detectable miRNA in vitro, the expression of miR-101b, miR-181b, miR-181d, miR-329, miR-3062 and miR-3065 was similar among different groups. However, the expression of miR-743a, miR-543, miR-101a, miR-1954, miR-712, miR-574-5p and miR-468 was higher in the arginine group compared with the control (*p* < 0.05). Furthermore, expression of miR-543, miR-101a, miR-1954, miR-712, miR-574-5p and miR-468 in the arginine group was higher compared with the control or casein groups (*p* < 0.05). The expression of miR-760-3p and miR-1954 was lowest in the casein group, compared with both control and arginine groups (*p* < 0.05). However, the expression of miR-875-3p was the lowest in the arginine group compared with both the casein and control groups. In addition, miR-875-3p in response to the casein group was lower compared to the control group.

### 3.2. Effects of Arginine Infusion on Milk Production in Lactating Cows

The milk protein concentration in the arginine group was greater compared to the casein or alanine groups (*p* < 0.05) (Table 7). Compared with the control (*p* < 0.05), milk and milk protein yield were higher in the arginine and alanine group. Compared with the casein group, the concentration of α-casein was higher in the arginine (*p* < 0.05) and alanine (*p* < 0.05) groups. No difference was observed in β-casein concentrations among treatments (*p* > 0.05). Compared with the alanine and the casein group, the concentration of κ-casein was the highest with arginine (*p* < 0.05). The total concentration of casein in milk (α-, β-, and κ-casein) and its daily yield were greater with the arginine compared with the alanine and casein groups (*p* < 0.05). The proportion of casein protein to milk protein was in the range of 73.18% to 92.63%, and greater in the arginine group, with 92.63% compared with the casein group. The mRNA expression of CSN1S1 and CSN1S2 was lower in the casein and alanine groups compared with the arginine group (*p* < 0.05). However, no difference was observed in the expression of CSN2 and CSN3 among treatments.

### 3.3. Effects of Arginine Infusion on Expression of mRNA and miRNA Involved in Milk Production

The mRNA expression of JAK2, STAT5, mTOR and S6K was greater in cows infused with arginine (*p* < 0.05) (Figure 1), but the mRNA expression of 4EBP1 was lower (*p* < 0.05) compared with the casein group. As described in Figure 2, the expression of miR-743a, miR-543, miR-101a and miR-760-3p in response to arginine infusion was greater compared with alanine or casein groups (*p* < 0.05). The expression of miR-1954 and miR-712 was greatest in the arginine group (*p* < 0.05). The expression of miR-574-5p in response to arginine and alanine infusion was similar, but greater than that in the casein group (*p* < 0.05). The expression of miR-468 was greatest in response to alanine (*p* < 0.05), and higher in the arginine compared with the casein group. 

## 4. Discussion

### 4.1. Effects of Arginine Infusion on Milk Protein production

The casein gene is a cluster composed of CSN1S1, CSN1S2, CSN2, and CSN3, that encode the αs1-, αs2-, β-, and κ-casein proteins, respectively. The αs1-casein and β-casein are the main casein types in milk. Although the content of κ-casein is low, it is an important component of milk casein. The proportion of α- casein, β- casein and κ-casein in the present study varied from 52.70% to 62.96%, 27.96% to 38.56% and 9.01% to 10.24%, respectively, which was similar to previous data [21]. Arginine plays an important role in casein synthesis in BMEC, by regulating the transcription of casein genes [7,22]. A study in lactating cows also found that arginine supplementation increased both milk yield and milk protein yield [23]. Although in the present study arginine supplementation did not alter the content of β-casein, which differed from our previous in vitro results in mammary epithelial cells [7], the observed responses for α-casein and the mRNA expression of CSN1S1 and CSN1S2 agreed with previous in vitro work with bovine mammary cells [7] and rat mammary tissue [24]. The increase in κ-casein content, as well as CSN3 expression in response to arginine, was consistent with the previous result [22]. The discrepancies among studies may be due to the inherent differences in metabolism between in vivo and in vitro systems [25], along with potential species-specific response [21,26].

Unlike data from the present study, enhanced EAA supply increased milk protein yield, but had no effects on the expression of csn2 in the mammary tissue of lactating cows [27]. Additionally, in the review by Cant et al. [28] evidence indicated that shifts in milk protein yield were not always accompanied by changes in the mammary expression of csn1, csn2, or csn3. Authors suggested that NE_L_ source and supply do not generally affect the expression of milk protein genes, but impact milk protein synthesis primarily through other mechanisms. The discrepancies between previous studies described and the present work may be attributed to specific mechanisms induced by EAA or energy availability, compared with functional amino acids such as arginine. Clearly, the regulation of casein and milk synthesis likely occurs through multifaceted mechanisms that response to the availability of EAA, cellular energy, and functional amino acids.

### 4.2. miRNA Regulation of Casein Gene Expression in Response to Arginine Supplementation

Shi and Gibson [29] reported that the up-regulation of mitochondrial malate dehydrogenase (MDH) by oxidative stress was mediated by miR-743a in mouse hippocampal cells. Therefore, we speculate that the up-regulation of miR-743a in response to arginine supply in the present study might have increased casein yield by inhibiting MDH activity. Because arginine has a protective effect on oxidative stress in rodents [30], it could also encompass the maintenance of a normal non-stress metabolic state in bovine mammary cells.

A previous study in BMEC showed that proliferation and casein protein concentration were increased by arginine [7,22]. A study in Wistar rats demonstrated an increase of both mammary acinar area and β-casein content in response to the dietary supply of arginine [24]. Similarly, Tan et al. [31] reported that 0.3 mM arginine in the medium was more suitable for intestinal epithelial cell proliferation and protein turnover. Together, the above data indicate that protein synthesis might be influenced by cell proliferation. miR-543 targets and inhibits SIRT1, a class III histone deacetylase, and promotes gastric cancer cell proliferation and cell cycle progression [32], suggesting that this miRNA plays a role in cell proliferation. In the context of arginine supply, the up-regulation of miR-543 expression and casein yield might have been associated with the stimulation of cell proliferation. It is generally believed that the dysregulation of miR-543 plays crucial roles in various human cancers, not only to promote cell proliferation and carcinoma [33], but also to inhibit cancer [34]. Based on data from the present study, however, miR-543 appears to be associated with the former.

Both proline dehydrogenase (PRODH) and ornithine aminotransferase (OAT) are the predicted target genes for miR-760-3p and miR-1954 (Table 2), which agrees with the greater expression of miR-760-3p and miR-1954 in response to arginine, both in vivo and in vitro. Thus, we speculate that the negative effect of miR-760-3p and miR-1954 on the proline pathway could help channel arginine through the polyamine pathway. As discussed above, it is possible that arginine enhances mammary cell proliferation and casein yield via polyamines [35] produced through the arginine-polyamine pathway.

miR-574-5p is a candidate oncogene in various types of cancers, such as colorectal cancer [36] and thyroid cancer [37,38]. miR-712 is generally believed to induce vascular endothelial inflammation and atherosclerosis [39], and has been shown to cause aortic aneurysms by stimulating matrix metalloproteinase activity [40]. In this study, the greater expression of miR-574-5p and miR-712 in response to arginine supply suggested that its effect on mammary cells could be related to the promotion of mammary cell proliferation, thereby further inducing casein synthesis in this group. However, the mechanisms by which they are linked to the mammary gland and lactation require further study.

The overexpression of miR-101a inhibited β-casein synthesis in mouse mammary cells (HC11) [12] suggesting this miRNA would be expected to suppress casein synthesis at least in part through targeting mTOR and ARGⅡ (Table 2). A previous study with Wistar rats showed that arginine supplementation in the diet increased mammary acinar area and β-casein content [24]. However, the abundance of α-casein and κ-casein in mammary gland tissue was not affected. Because in mouse mammary cells the overexpression of miR-101a also inhibits the cyclooxygenase 2 (COX-2)—prostaglandin pathway, it could be possible that more than one regulatory mechanism for casein synthesis encompassing the action of miR-101a exists. The fact that in a previous study [7], mTOR expression was also increased in response to arginine, seems to suggest that miR-101a up-regulation can play the same role in casein synthesis in the bovine as in the mouse mammary gland. This idea is supported by the fact that the increase of total casein protein yield in response to arginine was mainly derived from increases of α-casein and κ-casein, but not β-casein. We speculate that one reason for the impact of arginine on casein protein might be that this amino acid can regulate α-casein and κ-casein in bovine through a different mechanism than the one for β-casein reported in the mouse. Further research will have to be conducted to clarify the underlying mechanisms.

The present in vivo work showed that arginine infusion increased the content of α-casein and κ-casein in milk and the expression of CSN1S1 and CSN1S2, both of which are beneficial to the improvement of milk protein content. In addition, both in vitro and in vivo studies indicated that the differential expression of miR-743a, miR-543, miR-101a, miR-760-3p, miR-1954, and miR-712 in response to arginine supply might be a coordinated response that culminates in the regulation of casein synthesis. Clearly, physiological regulation in vivo is complex and in vitro studies cannot fully mimic the in vivo events. Thus, the precise in vivo regulatory mechanisms involving miRNA in the context of amino acid supply to the mammary gland merit further research.

## Figures and Tables

**Figure 1 animals-10-00797-f001:**
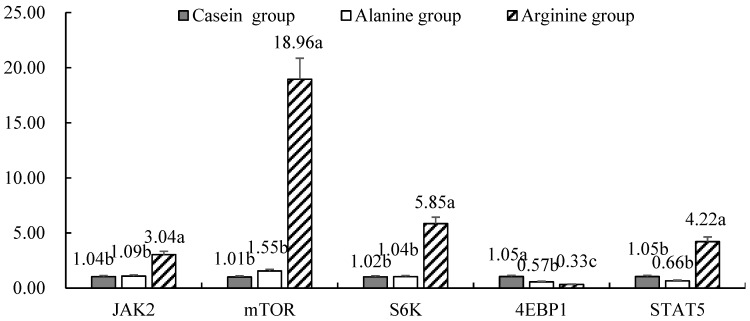
Expression of functional genes regulating casein yield in cow mammary tissue in the in vivo trial. Values with lower case letters were not different (*p* > 0.05). Values with different lower case letters differed (*p* < 0.05). JAK2 = Janus kinase 2; mTOR = Mechanistic target of rapamycin; S6K = Ribosomal protein S6 kinase; 4EBP1 = eIF4E-binding protein 1; STAT5 = Signal transducer and activator of transcription 5.

**Figure 2 animals-10-00797-f002:**
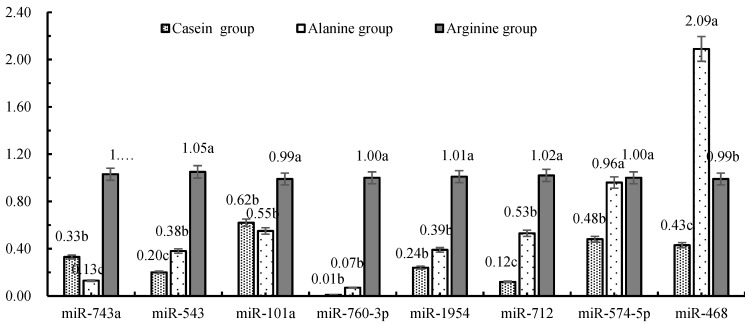
Expression of target miRNA in mammary tissue from lactating Holstein cows in response to jugular infusions. Values with different lower case letters differed (*p* < 0.05).

**Table 1 animals-10-00797-t001:** Composition of amino acids in culture media of bovine mammary epithelial cells (mM).

Amino Acids	Control Group	Casein Group	Arginine Group
Tyr	3.25	3.25	3.25
Ala	2.76	2.76	2.76
Gly	0.57	0.57	0.57
Glu	15.94	15.94	15.94
Ser	5.09	5.09	5.09
Cys	0.35	0.35	0.35
Phe	3.57	3.57	3.57
Leu	11.75	11.75	11.75
Ile	4.24	4.24	4.24
His	1.38	1.38	1.38
Lys	4.39	4.39	4.39
Thr	4.14	4.14	4.14
Met	2.22	2.22	2.22
Trp	1.05	1.05	1.05
Val	4.84	4.84	4.84
Pro	7.53	7.53	7.53
Asp	3.38	3.38	3.38
Arg	0	1.60	3.20

**Table 2 animals-10-00797-t002:** Candidate miRNA in various genes related to casein protein yield in dairy cows.

miRNAs	Target Gene Symbol			
miR-325	CSN1S1	CSN3	PRODH	ASL
miR-3062	CSN1S1	CSN3		
miR-471-3p	CSN1S1	ARGⅡ		
miR-186	CSN1S1	CSN3		
miR-465c-5p	CSN1S1	OAT		
miR-465b-5p	CSN1S1	OAT		
miR-465a-5p	CSN1S1	OAT		
miR-1942	CSN1S1	ARGⅡ		
miR-743b-3p	CSN1S1	OAT		
miR-743a	CSN1S1	OAT		
miR-543	CSN1S1	ARGⅡ	OAT	
miR-362-3p	CSN1S1	OAT		
miR-329	CSN1S1	OAT		
miR-694	CSN1S1	ARGⅠ	OAT	
miR-145	CSN2			
miR-335-3p	CSN3	ARG1		
miR-208a-5p	CSN3	OAT		
miR-3095-5p	CSN3	OAT		
miR-875-3p	CSN3	OAT		
miR-669b	CSN3	ARG1		
miR-3071	CSN3	ARGⅡ		
miR-574-5p	CSN3	PRODH		
miR-181d	CSN3	OAT		
miR-181a	CSN3	OAT		
miR-181b	CSN3	OAT		
miR-181c	CSN3	OAT		
miR-434-3p	CSN3	ARGⅠ		
miR-3067	CSN3	ARGⅡ		
miR-374	CSN3	OAT		
miR-384-3p	CSN3	ARGⅠ	OAT	PRODH
miR-410	CSN3	OAT		
miR-344d	CSN3	OAT		
miR-344e	CSN3	OAT		
miR-686	CSN3	ARGⅠ		
miR-141	CSN3	GLS		
miR-200a	CSN3	GLS		
miR-340-5p	CSN3	ARGⅠ		
miR-712	CSN3	ARGⅡ		
miR-205	CSN3	ARGⅡ		
miR-467f	CSN3	ARGⅠ		
miR-669m-3p	CSN3	ARGⅠ		
miR-409-3p	CSN3	ARGⅡ		
miR-101b	mTOR	ARGⅡ		
miR-101a	mTOR	ARGⅡ		
miR-466c-3p	ARGⅠ	OAT		
miR-466p-3p	ARGⅠ	OAT		
miR-466b-3p	ARGⅠ	OAT		
miR-532-5p	ARGⅡ	OAT		
miR-3065	ARGⅡ	OAT		
miR-5098	ARGⅡ	OAT		
miR-468	ARGⅡ	OAT		
miR-760-3p	OAT	PRODH		
miR-1954	OAT	PRODH		

CSN1S1 = αs1 - casein; CSN2 = β - casein; CSN3 = κ -casein; PRODH = proline dehydrogenase; OAT = Ornithine aminotransferase; ASL = argininosuccinate lyase; ARGⅠ = Arginase Ⅰ; ARGⅡ = Arginase Ⅱ; mTOR = Mechanistic target of rapamycin; GLS = Glutaminase.

**Table 3 animals-10-00797-t003:** Composition and nutrient levels of basal diets (DM basis).

Ingredients	Percentage/%	Nutrients	Level/%
Alfalfa	15.30	NE_L_/(MJ/kg) ^2)^	4.66
Chinese wildrye	10.47	CP	14.08
Silage	28.80	NFC	40.27
Corn	21.50	NDF	34.99
Cottonseed meal	6.10	ADF	21.09
Soybean meal	6.80	EE	3.96
Distillers Dried Grains with Soluble	9.40	Ca	0.91
CaHPO_4_	0.60	Total P	0.59
NaCl	0.50		
Premix^1)^	0.53		
Total	100.00		

^1^ The premix provided following per kg of diet: CuSO_4_ 25 mg, FeSO_4_·H_2_O 75 mg, ZnSO_4_·H_2_O 105 mg, Co 0.0024 mg, Na_2_SeO_3_ 0.016 mg, Vitamin A 12,000 IU, Vitamin D_3_ 10,000 IU, Vitamin E 25 mg, Nicotinic acid 36 mg, Choline 1000 mg. ^2^ NE_L_ in the diet was calculated according to the NE_L_ of ingredients and their percentages; concentrations of the other nutrients were measured values.

**Table 4 animals-10-00797-t004:** Composition of the treatment perfusates (g/L).

Amino Acids	Casein Group	Arginine Group	Alanine Group
Lys	21.50	21.50	86.00
Met	13.92	13.92	13.92
Phe	12.88	12.88	12.88
Ile	4.75	4.75	4.75
Arg	0.00	9.42	0.00
Ala	0.00	0.00	19.31

**Table 5 animals-10-00797-t005:** Primers for real-time PCR analysis of target genes and casein in bovine mammary epithelial cells.

Gene	Accession No.	Gene Description	Sequence (5′ → 3′)		Sources
*CSN1S1*	BC109618	αs1-casein	F	TCA CAG TAT GAA AGA GGG AA	Bos taurus
			R	AGC CAA TAG GAT TAG GGA	
*CSN1S2*	NM_174528.2	αs2-casein	F	AGG AAC GCA AAT GAA GAG	Bos taurus
			R	GGA GTA ATG GGA ACA GCA	
*CSN2*	NM_181008	β-casein	F	TGA GGA ACA GCA GCA AAC	Bos taurus
			R	CAG AGG CAG AGG AAG GTG	
*CSN3*	NM_174294	κ-casein	F	CGC TGT GAG AAA GAT GAA	Bos taurus
			R	AGA CCG CAG TTG AAG TAA	
*mTOR*	XM_001788228.1	Mechanistic Target of Rapamycin	F	CAT GTG CGA ACA CAG CAA CA	Bos taurus
			R	TGC CTT TCA CGT TCC TCT CC	
*JAK2*	XM_002689603.1	Janus kinase 2	F	ACA GGG ATC TGG CAA CAA GG	Bos taurus
			R	CGC ATA AAT TCC GCT GGT GG	
*STAT5*	NM_001012673.1	Signal transducer and activator of transcription 5	F	CAA TGG ACA GTC TGG AGC CC	Bos taurus
			R	CCT GCA CAC TGG GGA TTG TT	
*S6K*	NM_205816.1	Ribosomal protein S6 kinase	F	CGG AAC AGT CAC ACA CAC CT	Bos taurus
			R	TGG CTT CTT GCG TGA GGT AG	
*4EBP1*	BC120290.1	EIF4E binding protein1	F	CGG AAC TCA CCT GTG ACC AA	Bos taurus
			R	AGG TGA TTC TGC CTG GCT TC	
*GAPDH*	XM_001252479.1	Glyceraldehyde-3-phosphate dehydrogenase	F	CCC CGC GCT CTA ATG TTC A	Bos taurus
			R	GAA GGG GTC ATT GAT GGC GA	
*ACTB*	NM_173979.3	β-actin	F	ACT GTT AGC TGC GTT ACA CCC TT	Bos taurus
			R	TGC TGT CAC CTT CAC CGT TCC	

Note: All the primers used in this experiment were synthesized in Invitrogen (Nanjing, China).

**Table 6 animals-10-00797-t006:** Expression of miRNA screened from PicTar and TargetScan databases in bovine mammary epithelial cells in vitro.

miRNAs	Treatments			SEM	*p*-Value
	Control	Casein	Arginine		
miR-101b	1.32	1.28	0.98	0.11	0.342
miR-181b	0.82	0.93	0.84	0.063	0.749
miR-181d	1.23	1.03	1.22	0.083	0.792
miR-329	1.79	2.14	1.96	0.189	0.993
miR-3062	2.27	1.74	2.47	0.17	0.463
miR-3065	0.90	1.16	1.02	0.083	0.212
miR-743a	1.20^b^	2.10^ab^	2.73^a^	0.213	<0.001
miR-543	0.68^b^	0.61^b^	1.60^a^	0.103	0.012
miR-101a	0.22^b^	0.37^b^	3.99^a^	0.157	0.003
miR-760-3p	1.76^a^	0.82^b^	2.35^a^	0.127	0.034
miR-1954	1.99^b^	0.7^c^	5.63^a^	0.183	<0.001
miR-712	0.42^b^	0.52^b^	2.44^a^	0.147	0.01
miR-574-5p	0.95^b^	0.35^b^	5.63^a^	0.107	<0.001
miR-468	0.23^b^	0.30^b^	1.48^a^	0.047	0.041
miR-875-3p	48.26^a^	17.17^b^	2.97^c^	1.653	0.001

^a,b^ Values with different letters in the same row differed significantly (*p* < 0.05).

**Table 7 animals-10-00797-t007:** Effect of jugular vein arginine infusion on the yield of milk protein and milk casein protein in Holstein cows.

Items	Treatments			SEM	*p*-Value
	Casein	Alanine	Arginine		
Average daily intake/(kg)	22.82	21.64	22.3	1.625	0.461
Milk protein/%	3.04^b^	3.11^b^	3.17^a^	0.077	0.046
Milk yield/(kg/d)	21.45^b^	23.65^a^	24.16^a^	0.823	0.039
Milk protein yield/(kg/d)	0.65^b^	0.74^a^	0.77^a^	0.027	0.04
α-Casein/(g/L)	11.70^b^	16.37^a^	17.24^a^	1.417	0.008
β-Casein/(g/L)	8.56	7.27	9.09	0.523	0.985
κ-Casein/(g/L)	2.00^b^	2.37^b^	3.00^a^	0.194	0.024
Casein content/%	2.23^c^	2.60^b^	2.93^a^	0.167	0.031
Casein yield/(kg/d)	0.48^c^	0.62^b^	0.71^a^	0.053	0.044
Casein/milk protein/%	73.18^b^	83.85^ab^	92.63^a^	6.613	0.003
αs1-casein	0.96^b^	1.00^b^	1.54^a^	0.113	0.028
αs2-casein	0.98^b^	1.03^b^	1.39^a^	0.137	0.037
β-casein	1.03	1.43	1.22	0.117	0.832
κ-casein	1.00	1.03	1.36	0.103	0.327

^a,b^ Values with different letters in the same row differed significantly (*p* < 0.05).

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
