# Peer review of "Arginine Supply Impacts the Expression of Candidate microRNA Controlling Milk Casein Yield in Bovine Mammary Tissue"

_animals, 2020, doi:10.3390/ani10050797_

Round 1
Reviewer 1 Report
Authors address all the comments clearly.Author Response
Authors address all the comments clearly.
Response: Thanks for your comment.
Reviewer 2 Report
It is unclear why arginine and alanine give same results. Please explain. I remember to authors that microRNAs are more complex, they should add table and figure explaining their results and commenting importance not only for bovines but for humans.
Author Response
It is unclear why arginine and alanine give same results. Please explain. I remember to authors that microRNAs are more complex, they should add table and figure explaining their results and commenting importance not only for bovines but for humans.
Response: Thanks for comments. In present study, alanine was used as iso-nitrogenous control to arginine in order to get rid of the effect of nitrogen amount by Arginine infusion, and aimed to investigate the effect of arginine infusion on milk protein yield and nitrogen utilization, as alanine is considered a NEAA and non-function AA. However, the daily production of α-casein in the cows infused with alanine or arginine were both higher than that in control group. We presume to attribute the effects of alanine on α-casein synthesis to its function as a glucogenic amino acid. As reported by previous study alanine can be readily converted to glucose with the help of catalytic action of glutamate-pyruvate transaminase in liver (Litwack and Litwack, 2018). The liver glucose can then be released to provide energy to other organs like muscle and mammary gland, resulting in the increased milk yield together with α-casein production. In addition, we supplemented some literature about related miRNA( page15, line 366-383), mainly on examples of miRNAs in cell proliferation and cancer, with relatively single types. At present, there are still few studies on the effect of miRNA on lactation, and the relevant literature in this study is limited. However, it is our new findings that broaden this field and point out the direction for the study of related mechanisms.
Reviewer 3 Report
1. During the experiment (table 1), the basis of adding arginine concentration in cells.
2. Methods for predicting candidate mirnas involving casein yield are detailed in the materials method.
3. Results 3.3 it is suggested to supplement the protein expression of JAK2, mTOR, S6K, 4EBP1 and STAT5
Author Response
1. During the experiment (table 1), the basis of adding arginine concentration in cells.
Response 1: Thanks for the comments. The basic medium used for cell culturing was made by ourselves with the composition listed in the part 2.1.1 in manuscript (according to the composition of DMEM/F12, Gibco, Invitrogen, Catalog #11320082, Life Technologies Corporation, US).
2. Methods for predicting candidate mirnas involving casein yield are detailed in the materials method.
Response 2: Thanks for the comments. We have added relevant content as you suggested ( Page 4, Line 131 ).
3. Results 3.3 it is suggested to supplement the protein expression of JAK2, mTOR, S6K, 4EBP1 and STAT5
Response 3: This is a constructive suggestion, which is very helpful for the scientific and integrity of the paper. However, the changes genetic expression would also be beneficial to indicate the possible mechanism for the regulation on protein synthesis by Arg injection. We appreciate for your suggestion while it might be hard to re-run the test for protein expression. Because the samples were collected for genetic expression test which is not enough for any more test.
This manuscript is a resubmission of an earlier submission. The following is a list of the peer review reports and author responses from that submission.
Round 1
Reviewer 1 Report
The article describes two experiments designed to provide information on the role of arginine in controlling milk casein yield in dairy cattle. One is an in vitro experiment and one is an in vivo experiment. The in vitro studies demonstrate a differential response in specific microRNA to treatments that differed in arginine concentration. The in vivo study infused treatments containing different arginine concentration but similar N concentration. There were differential responses in milk components to treatments.
There are numerous studies supporting the role of arginine as a functional amino acid. The potential role of arginine in regulating milk casein yield is of interest. The focus on microRNA is a novel area.
The paper is not well written and in need of extensive editing for proper word use and sentence structure. References to previous literature are misleading in several cases. For example, reference 1 (Kim et al., 2004) describes a study with piglets that were fed diets deficient or adequate in the amount of arginine. The study showed that addition of arginine to diets that were deficient resulted in increased growth. The Kim et al. article has 3 authors; only 2 authors are listed. Reference 2 (Doepel and Lapierre, 2011) included abomasal infusion of 3 treatments – water, a mixture of essential AA without Arg, and a mixture of essential AA with Arg. The response in milk yield and milk protein yield increased with both amino acid treatments. There was no difference between the infusion with or without Arg.
In a previous experiment reported from the lab of the authors (Ding et al., JDS 101:3514-3523, 2018) “Six Chinese Holstein lactating cows with similar BW (550.0 ± 20 kg), parity (4), BCS (3.0), milk yield (21.0 ± 1.0 kg), and DIM (80 ± 2 d) were selected for this experiment. ….Cows were randomly assigned to 3 treatments in a replicated 3 × 3 Latin square design with 22 d for each period (d 1 to 7 for infusion and d 8 to 22 for washout)”
The animals in the current study are identical to the ones in Ding et al. “Six Holstein lactating cows with similar BW (mean ± SE) (550.0 ± 20 kg), parity (4), BCS (3.0 ± 0.0), milk yield (21.0 ± 1.0 kg), and DIM (80 ± 2 d) were selected for this experiment…” “Experimental animals were randomly divided into 3 treatments (2 cows per group) in a replicated 3×3 Latin square design with 22-day for each period (day 1 to day 7 for infusion and day 8 to day 22 for washout). It is highly unlikely that two groups of 6 lactating cows would be identical (mean and SE) in BW and milk yield at the same DIM. Perhaps the authors listed the wrong cows for the current experiment. This is concerning relative to evaluating the data in the manuscript.
The authors are infusing 37.66 g of Arg over 8 hr. Arg is a known secretagogue. There are no data in the manuscript on concentrations of Arg following infusion. How much was Arg concentration increased? Were concentrations high enough to promote secretion of growth hormone or other hormones that could contribute to the response? These data are needed to interpret the results of the experiment.
Line 76 – define the casein genes
Table 2 – define target gene symbols; clarify the source of candidate miRNAs
Line 163 – include composition of the treatment infusates
Line 164 – Arginine infused was 37.66 g over 8 hr.
Table 4 – define the genes
Table 6 define terms and units for last 4 items in table
Figure 1 – define genes
Line 270 – incorrect reference listed
Reviewer 2 Report
Page 4 line 122-132, What’s the object of this section? Which miRNAs will be qualified? Page 4 line 132, "The other primer sequences of the target miRNA are reported in Table 2". There isn’t any information of primer sequences in Table 2. Page 4 line 134, Whether PicTar could be used to predict miRNAs target genes in bovine? Page 9 line 201, The formula for calculating ΔΔCt is not clear. Page 10 line 230-231, "Compared with the casein group, the concentration of α-casein was higher in the arginine (P < 0.05) and alanine (P < 0.05) groups". Why was alanine group higher than casein group? Page 14 line 332-338, PRODH and OAT are predicted target genes for miR-760-3p and miR-1954 (Table 2),why not quantify the mRNA expression of the two genes through experimental method?7.MiRNAs in this study were selected by prediction (Candidate miRNAs in various genes related to casein protein yield of dairy cows, Table2), but the authors’s results suggested these miRNAs were not the targets. What's the significance of choosing these miRNAs to study?
Most of miRNAs delected in this study were up-regulated. The candidate target genes that related to milk protein production were also up-regulated. Neither of them is negatively correlated. It's contradictory.